# META-ACTIVE LEARNING IN PROBABILISTICALLY-SAFE OPTIMIZATION

## ABSTRACT

Learning to control a safety-critical system with latent dynamics (e.g. for deep brain stimulation) requires judiciously taking calculated risks to gain information. We present a probabilistically-safe, meta-active learning approach to efficiently learn system dynamics and optimal configurations. The key to our approach is a novel integration of meta-learning and chance-constrained optimization in which we 1) meta-learn an LSTM-based embedding of the active learning sample history, 2) encode a deep learning-based acquisition function with this embedding into a mixed-integer linear program (MILP), and 3) solve the MILP to find the optimal action trajectory, trading off the predicted information gain from the acquisition function and the likelihood of safe control. We set a new state-of-the-art in active learning to control a high-dimensional system with latent dynamics, achieving a 46% increase in information gain and a 20% speedup in computation time. We then outperform baseline methods in learning the optimal parameter settings for deep brain stimulation in rats to enhance the rats' performance on a cognitive task while safely avoiding unwanted side effects (i.e., triggering seizures).

## 1 INTRODUCTION

Safe and efficient control of a novel systems with latent dynamics is an important objective in domains from healthcare to robotics. In healthcare, deep brain stimulation devices implanted in the brain can improve memory deficits in patients with Alzheimers (Posporelis et al., 2018) and responsive neurostimulators (RNS) can counter epileptiform activity to mitigate seizures. Yet, the surgeon's trial-and-error process of finding effective RNS parameters for each patient is time-consuming and risky, with poor device settings possibly damaging the brain.

Researchers studying *active learning* and *Bayesian optimization* have sought to develop algorithms to efficiently and safely learn a systems' dynamics, e.g. learning a brain's dynamics for RNS configuration (Ashmaig et al., 2018; Sui et al., 2018). However, because these algorithms fail to scale up to higher-dimensional state-action spaces, researchers utilize only simple voltage and frequency controls rather than all 32 channels of the RNS waveform (Ashmaig et al., 2018). Similarly, tasks in robotics, e.g. learning the dynamics of novel robotic systems (e.g., an autopilot learning to fly a damaged aircraft), require active learning methods that succeed in higher-dimensional domains.

In this paper, we develop a probabilistically-safe, meta-active learning approach to tackle these challenging tasks to efficiently learn system dynamics and optimal configurations. We draw inspiration from recent contributions in meta-learning (Finn et al., 2017; Nagabandi et al., 2019; Wang et al., 2016; Andrychowicz et al., 2016) that seek to leverage a distribution over training tasks to optimize the parameters of a neural network for efficient, online adaptation. Researchers have previously investigated meta-learning for active learning, e.g. learning a Bayesian prior over a Gaussian Process (Wang et al., 2018b) for learning an acquisition function. However, these approaches do not consider the important problem of safely and actively learning to control a system with altered dynamics, which is a requirement for safety-critical robotic applications. Furthermore, as we show in Section 5, on challenging control tasks for healthcare and robotics, the performance of prior active learning approaches (Kirsch et al., 2019; Hastie et al., 2017) leaves much to be desired.

We seek to overcome these key limitations of prior work by harnessing the power of meta-learning for active learning in a chance-constrained optimization framework for safe, online adaptation by encoding a learned representation of sample history. Instead of hand-engineering an acquisition

function for our specific domains, our approach employs a data-drive, meta-learning approach, which results in better performance than prior approaches, as shown in Section 5. Furthermore, our approach has the unique ability to impose analytical safety constraints over a sample trajectory.

**Contributions –** We develop a probabilistically safe, meta-learning approach for active learning ("meta-active learning") that sets a new state-of-the-art. Our acquisition function (i.e., the function that predicts the expected information gain of a data point) is meta-learned offline, allowing the policy to benefit from past experience and provide a more robust measure of the value of an action. The key to our approach is a novel interweaving of our deep, meta-learned acquisition function as a Long-Short Term Memory Network (Gers et al., 1999) (LSTM) within a chance-constrained, mixed-integer linear program (MILP) (Schrijver, 1998). By encoding the LSTM's linear, piece-wise output layers into the MILP, we directly optimize an action trajectory that best ensures the safety of the system while also maximizing the information learned about the system. In this paper, we describe our novel architecture which uniquely combines the power of a learned acquisition function with chance-constrained optimization and evaluate its performance against state-of-the-art baselines in several relevant domains. To the best of our knowledge, this is the only architecture which meta-learns an acquisition function for optimization tasks and is capable of embedding this acquisition function in a chance-constrained linear program to guarantee a minimum level of safe operation.

The contributions of this paper are as follows:
1. Meta-active learning for autonomously synthesizing an acquisition function to efficiently infer altered or unknown system dynamics and optimize system parameters.
2. Probabilistically-safe control combined with an active-learning framework through the integration of our deep learning-based acquisition function and integer linear programming.
3. State-of-the art results for safe, active learning. We achieve a 46% increase in information gain in a high-dimensional environment of controlling a damaged aircraft, and we achieve a 58% increase in information gain in our deep brain stimulation against our baselines.

## 2 PRELIMINARIES

In this section, we review the foundations of our work in active, meta-, and reinforcement learning.

**Active Learning** – Labelled training data is often difficult to obtain due either to tight time constraints or lack of expert resources. Active learning attempts to address this problem by utilizing an "acquisition function" to quantify the amount of information an unlabelled training sample, $x \in D_U = \langle x_i \rangle_{i=1}^n$, would provide a base learner, $\hat{T}_\psi$, if that sample were given a label, $y$ and added to a labeled dataset, $D_L = \langle x_j, y_j \rangle_{j=1}^m$, i.e., $D_L \leftarrow D_L \cup \langle x, y \rangle$. The active learning algorithm queries its acquisition function, $H(D_U, D_L, T_\psi)$, to select which $x \in D_U$ should be labeled and added to $D_L$; then, a label is queried (e.g., by taking an action in an environment and observing the effect) for $x$, and the new labeled sample is added to $D_L$ (Muslea et al., 2006; Pang et al., 2018).

**Meta-Learning** – Meta-learning approaches attempt to learn a method to quickly adapt to new tasks online. In contrast to active learning, meta-learning attempts to learn a skill or learning method, e.g. learning an active learning function, which can be transferable to novel tasks or scenarios. These tasks or skills are trained offline, and a common assumption is that the tasks selected at test time are drawn from the same distribution used for training (Hospedales et al., 2020).

**Reinforcement Learning and Q-Learning** – A Markov decision process (MDP) is a stochastic control process for decision making and can be defined by the 5-tuple $\langle \mathcal{X}, \mathcal{U}, \mathcal{T}, \mathcal{R}, \gamma \rangle$. $\mathcal{X}$ represents the set of states and $\mathcal{U}$ the set of actions. $T : \mathcal{X} \times \mathcal{U} \times \mathcal{X}' \rightarrow [0, 1]$ is the transition function that returns the probability of transitioning to state $x'$ from state $x$ applying action, $u$. $\mathcal{R} : \mathcal{X} \times \mathcal{U} \rightarrow \mathbb{R}$ is a reward function that maps a state and action to a reward, and $\gamma$ weights the discounting of future rewards. Reinforcement learning seeks to synthesize a policy, $\pi : \mathcal{X} \rightarrow \mathcal{U}$, mapping states to actions to maximize the future expected reward. When $\pi$ is the optimal policy, $\pi^*$, the following Bellman condition holds: $Q^{\pi^*}(x, u) := \mathbb{E}_{x' \sim T} \left[ R(x, u) + \gamma Q^{\pi^*}(x', \pi^*(x)) \right]$ (Sutton & Barto, 2018).

### 2.1 PROBLEM SET-UP

Our work is at the unique nexus of active learning, meta-learning and deep reinforcement learning with the objective of learning the Q-function as an acquisition function to describe the expected future information gained when taking action, $u$, in state, $x$, given a set of previously experienced states

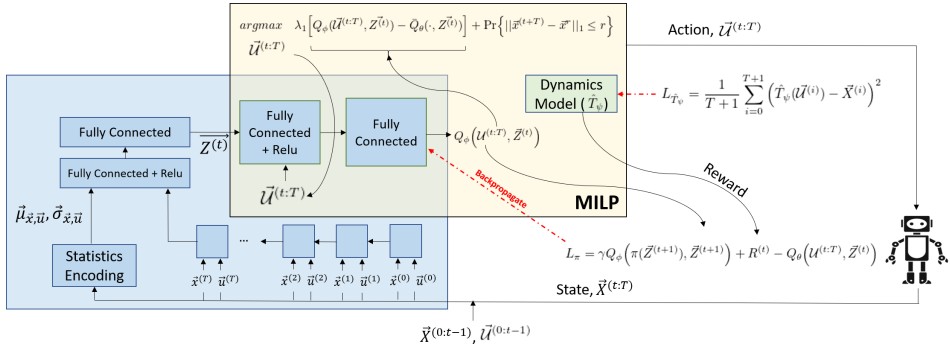

Figure 1: This figure depicts our framework. Previously collected state-action pairs are fed into an LSTM embedding layer. Mean and variance statistics are also calculated for each state and action. The concatenated vector is fed through a fully connected layer to create the $\vec{Z}$ embedding. Our Q-function consists of two, fully-connected, ReLU layers. All blue elements are neural network layers of our acquisition function; the green element is the dynamics model; and the components of the MILP are housed in the yellow box.

and actions. We define information gain as the percent decrease in the error of the objective (e.g., decrease in model error). A formal definition is provided in the Appendix. The Q-function is trained via a meta-learning strategy in which an agent interacts in environments sampled from a distribution of different scenarios to distill the most effective acquisition function for active learning of system dynamics. In our context, the state, $\mathfrak{X}$, of the active learning system is given by $\mathfrak{X} : \langle D_U, D_L, \hat{T}_\psi \rangle$, where $D_U$ consists of all possible state-action pairs, $D_L$ is the set of state-action pairs that the agent has already experienced, and $\hat{T}_\psi$ is a neural network function approximator of the transition dynamics, which is parameterized by $\psi$ and updated online as samples are collected. We note that this state, $\mathfrak{X}$, is distinct from the state, $X$, of the robotic (or other) system we are controlling. The reward, $R^{(i)}$, is proportional to the reduction of the mean squared error (MSE) loss of $\hat{T}_\psi$ at time step, $i$.

## 3 SAFE META-LEARNING ARCHITECTURE

Several key components are vital for learning about an unknown system in a timely manner. First, an encoded representation of the context of the new dynamics is important for determining where exploration should be focused and which actions may be beneficial for gaining information of unknown dynamics. Second, a range of prior experiences in active learning should be leveraged to best inform which actions elicit the most information in a **novel context** within a distribution of tasks. We seek to develop a framework with these key attributes to enable sample-efficient and computationally light-weight active learning. An overview of our system is shown in Fig. 1.

### 3.1 META-LEARNING ALGORITHM

To infer the Q-function for an action (i.e. the acquisition function), we meta-learn over a distribution of altered dynamics as described in Algorithm 1. For each episode, we sample from this distribution of altered dynamics and limit each episode to the number of time steps, $M$, tuned to collect enough data to accurately learn our approximate dynamics, $\hat{T}_\psi$, as a neural network with parameters, $\psi$. We utilize Q-learning to infer the Q-function which is represented by a DQN. In our training scheme we search over the action space via a MILP, as described in Section 3.2 and select the action set, $\vec{\mathcal{U}}^{(t:T)}$, which maximizes the Q-function while satisfying safety constraints.

The acquisition Q-function, $Q_\theta$, as described in Eq. 1 is trained via Deep Q-Learning (Ganger et al., 2016) with target network, $Q_\phi$. During training, our Deep Q-Learning framework is augmented to account for safety . The learned acquisition function, $Q_\theta$, is utilized by our policy (Eq. 1), which is solved via a safety-constrained MILP solver. The reward, $R^{(t)}$, for taking a set of actions in a given state is defined as the percent decrease in the MSE error of the model, $\hat{T}_\psi$. This Bellman loss is backpropagated through the Q-value linear and (ReLU) output layers through the LSTM encoding

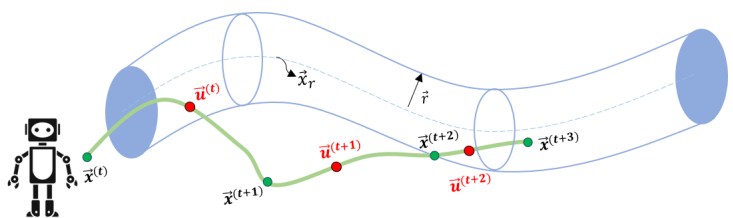

Figure 2: This figure depicts the d-dimensional hyper-ellipsoid of safety. $\vec{x}_r$ denotes the safest state for the system and the shaded region is the set of all safe states. Action, $\vec{u}^{(t)}$, is an exploratory action, which may bring the system outside of the hyper-ellipsoid of safety. Action, $\vec{u}^{(t+1)}$, estimates that the system returns to a safe state with probability $1 - \epsilon$.

layers. $\pi(\vec{Z}^{(t+1)})$ is the set of actions, $\vec{\mathcal{U}}^{(t+1)}$, determined by maximizing Eq. 1, which we describe in Section 3.2. The dynamics model, $\hat{T}_\psi$, is retrained with each new set of state-action pairs.

---

**Algorithm 1** Meta-learning for training

1: Randomly initialize $Q_\theta$ and $Q_\phi$ with weights $\theta = \phi$
2: Initialize replay buffer, D
3: **for** episode=1 to N **do**
4:     Initialize $\hat{T}_\psi$ based on meta-learning distribution
5:     Collect small initial set of state-action pairs, $\mathcal{U}^{(0)}, X^{(0)}$
6:     Train $\hat{T}_\psi$ on initial set
7:     **for** t=1 to M **do**
8:         Forward pass on encoder to obtain $\vec{Z}^{(t)}$
9:         Select $\mathcal{U}^{(t)}$ from Eq. 1
10:         Execute actions $\mathcal{U}^{(t)} + \mathcal{N}$ according to exploration noise, $\mathcal{N}$; observe states $X^{(t+1)}$
11:         Retrain $\hat{T}_\psi$ on $X^{(0:t)}, \mathcal{U}^{(0:t)}$ and observe reward $R^{(t)}$
12:         $D \leftarrow D \cup \langle \mathcal{U}^{(t)}, X^{(t)}, X^{(t+1)}, R^{(t)} \rangle$
13:         Sample a batch of transitions from D
14:         Perform forward pass through encoder to obtain $\vec{Z}^{(t+1)}$
15:         Calculate $y^{(t)} = R^{(t)} + \gamma Q_\phi\left(\pi(\vec{Z}^{(t+1)}), \vec{Z}^{(t+1)}\right)$
16:         Update $Q_\theta$ according to $\left(y^{(t)} - Q_\theta(\vec{\mathcal{U}}^{(t)}, \vec{Z}^{(t)})\right)$
17:         $Q_\phi \leftarrow \tau Q_\theta + \tau(1 - Q_\phi)$
18:     **end for**
19: **end for**;

**Algorithm 2** Meta-learning for testing

1: Draw test example from distribution
2: Initialize $\hat{T}_\psi$
3: **for** t=1 to M **do**
4:     Do forward pass through encoder to get $\vec{Z}^{(t)}$
5:     Select actions, $\mathcal{U}^{(t)}$, according to Eq. 1
6:     Execute actions, $\mathcal{U}^{(t)}$; observe states $X^{(t+1)}$ and reward $R^{(t)}$
7:     Retrain $\hat{T}_\psi$ on $X^{(0:t+1)}$, $\mathcal{U}^{(0:t+1)}$
8: **end for**

---

### 3.2 MIXED-INTEGER LINEAR PROGRAM

Our objective (Eq. 1) is to maximize both the probability of the system remaining in a safe configuration and the information gained along the finite trajectory horizon from $[t, t + T)$.

$$\vec{\mathcal{U}}^{(t:t+T)*} = \operatorname*{argmax}_{\vec{\mathcal{U}}^{(t:t+T)} \in \vec{\boldsymbol{\mathcal{U}}}^{(t:t+T)}} \lambda_1 \left[ Q_\phi(\vec{\mathcal{U}}^{(t:t+T)}, \vec{Z}) - \bar{Q}_\theta(\cdot, \vec{Z}) \right] + \Pr\left\{ \left\| \vec{x}^{(t+T)} - \vec{x}_r \right\|_1 \leq r \right\} \quad (1)$$

$Q_\phi(\vec{\mathcal{U}}^{(t:t+T)}, \vec{Z})$ describes the expected information gained along the trajectory when the set of actions $\vec{\mathcal{U}}^{(t:t+T)}$ is taken in the context of the embedding $\vec{Z}$, and $\bar{Q}_\theta(\cdot, \vec{Z})$ is the expected Q-value in state $\vec{Z}$, which we discuss further in Section 3.3. $\lambda_1$ is a hyper-parameter that can be tuned to adjust the tradeoff between safety and information gain. $\Pr\left\{ \left\| \vec{x}^{(t+T)} - \vec{x}_r \right\|_1 \leq r \right\}$ defines the probability of remaining within the hyper-ellipsoid of safety. We derive a linearization of this equation based on an assumption of Gaussian dynamics which can then be solved via MILP.

**Definition of Safety -** We define a hyper-ellipsoid of safety, parameterized by the safe state $x_r$ and radius $\vec{r}$ encompassing all known safe system states. In the case of an aircraft, $\vec{x}_r$ would be straight and level flight and the hyper-ellipsoid would include deviations considered safe. Additionally, we

assume that our model error comes from a Gaussian distribution with known mean and variance. This assumption allows us to impose safety constraints which can either be enforced with a probability of one, or this requirement can be relaxed to increase the potential for information gain.

By assuming that our model error originates from a Gaussian distribution, we can linearize our probability constraints described in Eq. 9 to include in the MILP (See Appendix for full derivation). Here, $\Phi^{-1}$ is the inverse cumulative distribution function for the standard normal distribution and $1 - \epsilon_d$ denotes the probability level. $\sigma$ represents the uncertainty in the dynamics network parameters. $d$ represents the $d^{th}$ row and $j$ represents the $j^{th}$ column. $\Delta_d^{(t:t+T)} = x_d^r - [\beta x^{(t)}]_d$ where $\beta$ defines the evolution of a state (See Appendix A.4.1 for more detail). We compute the uncertainty, $\sigma$, of our network via bootstrapping (Efron, 2020; Franke & Nuemann, 1998), which we describe further in the Appendix. We guarantee a minimum probability of safety by selecting the minimum risk value, $\epsilon_d$, to be at or above our minimum desired probability of safety. This procedure ensure that our algorithm must select a safety constraint that exceeds this minimum safety requirement. Note: the hyper-ellipse, when linearized, is a hyper-rectangle.

$$\Pr\left\{ \left\| \vec{x}^{(t+T)} - \vec{x}_r \right\|_1 \leq r \right\} \longrightarrow \left\| \Phi^{-1}(1-\epsilon_d)\sqrt{\sum_j \sigma_{d,j}^2 x_j^{(t)2} + \sum_j \sigma_{d,j}^2 \mathcal{U}_j^{(t:t+T)2}} + \Gamma_t \vec{\mathcal{U}}^{(t:t+T)2} - \Delta_d^{(t:t+T)} \right\|_1 < r_d, \forall d \tag{2}$$

Our policy takes a set of information rich actions at time $t$, potentially out of the d-dimensional hyper-ellipsoid of safety, guaranteeing with probability $1 - \epsilon$ that the system will return to a safe state after $T$ time steps. Thus, our $T$-step propagation allows the system to deviate from the safe region, if desired, to focus on actively learning so long as it can return to the safe region with high confidence.

### 3.3 AN LSTM Q-FUNCTION AS A LINEAR PROGRAM

We leverage an LSTM as a function approximator for Q-learning. To blend our Q-function with the MILP, we pass the LSTM output embedding, which serves as a learned representation of the sample history, through a fully connected layer with ReLU activations.[1] This design enables us to backprop the Bellman residual through the output layers encoded in the MILP all the way to the LSTM inputs. Thus we can leverage the power of an LSTM and mathematical programming in a seamless framework.[2] Given our meta-learned acquisition function and chance-constrained optimization framework, we can now perform probabilistically-safe active learning. Algorithm 2 describes how we perform our online, safe and active learning. Intuitively, our algorithm initializes a new dynamics model to represent the unknown or altered dynamics, and we iteratively sample information rich, safe actions via our MILP policy, update our dynamics model, and repeat.

## 4 EXPERIMENTAL EVALUATION

We design two experiments to validate our approach's ability to safely and actively learn altered or unknown dynamics in real time. We compare our approach to several baseline approaches and demonstrate that our approach is superior in terms of both information gain and computation time. More details on the experimental domains can be found in the Appendix.

### 4.1 ASYNCHRONOUS DISTRIBUTED MICROELECTRODE THETA

Asynchronous distributed microelectrode theta stimulation (ADMETS) is a cutting edge deep brain stimulation approach to treating seizure conditions that cannot be controlled via pharmacological methods. In ADMETS, a neuro-stimulator is implanted in the brain to deliver continuous electrical pulses to reduce seizures. However, there is no clear mapping from parameter values to reduction in seizures that applies to all patients, as the optimal parameter settings can depend on the placement of the device, the anatomy of an individual's brain and other confounding factors. Further, a latent subset of parameters can cause negative side-effects.

---

[1]We include the mean and standard deviation of previously collected states and actions as we find the first and second moments of the data to be helpful, additional features.

[2]Details on the hyperparameter settings including learning rates and layer sizes can be found in the Appendix along with the derivation for the linearization of the Q-function for inclusion in the linear program.

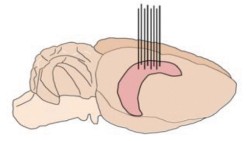
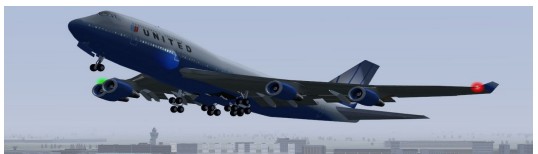

(a) ADMETS Implantation.    (b) High-dimensional Domain; Courtesy: Flightgear.

Figure 3: Fig. 3a depicts a surgical ADMETS device implanted in a rat brain (Ashmaig et al., 2018) (Section 4.1). Fig. 3b depicts the high-dimensional domain for dynamical control (Section 4.2).

As a surrogate task for evaluating how well an algorithm can find the optimal paremeters of an ADMETS device for in vivo epilepsy therapy, researchers have proposed leveraging experimental data collected from parameter sweeps of ADMETS devices in rats. In keeping with Ashmaig et al. (2018), we create simulation environments for six rats where, at each ADMETS parameter setting, the cognitive function of a rat was tested (i.e., the rats ability to recall where a treat was located in a maze), as measured by a "memory score." The task for safe, active learning task is to determine the ADMETS parameters (i.e., signal amplitude) in the simulation environments that maximize each rat's memory scores without causing unwanted side effects (e.g., seizures), which can arise when the memory score drops below zero. The reward signal utilized by our meta-learner is the percent decrease in error between the predicted and actual optimal parameters.

## 4.2 HIGH-DIMENSIONAL DOMAIN (RECOVERING DYNAMICS OF A DAMAGED AIRCRAFT)

Active learning algorithms can be ineffective in high-dimensional domains. As such, we seek to stress-test our algorithms in just such a domain: Learning the nonlinear dynamics of a damaged aircraft online before the system enters an unrecoverable configuration (e.g., a spin or crashing). We base our simulation on theoretical damage models from prior work describing the full equations of motion (Watkiss, 1994; Zhang et al., 2017; Ouellette, 2010) within the Flightgear virtual environment. The objective of this domain is to learn the difference between the altered dynamics that result from the damage and to maintain safe flight. We consider safe flight to be our designated safe state, $\vec{x}_r$. The aircraft takes an information rich action potentially resulting in a deviation outside of the d-dimensional hyper-ellipsoid of safety. The next action is constrained to guarantee that the plane returns to the hyper-ellipsoid of safety with probability $1 - \epsilon$ via action $u^{(t+1)}$.

## 4.3 BASELINE COMPARISONS

We evaluate against the following baselines in active learning and Bayesian optimization. The acquisition function baselines, Epistemic Uncertainty (Hastie et al., 2017) and Maximizing Diversity (Schrum & Gombolay, 2020) are embedded in our safety framework, therefore providing a head-to-head comparison between our meta-learned acquisition function and these active learning heuristics.

- **Epistemic Uncertainty (Hastie et al., 2017)** - This active learning metric selects the action that maximizes uncertainty.
- **Maximizing Diversity (Schrum & Gombolay, 2020)** - This acquisition function selects actions which maximize the difference between previously seen states and actions.
- **Bayesian Optimization (BaO) (Ashmaig et al., 2018)** - This algorithm was developed for the ADMETS domain (Section 4.1) and is based upon a Gaussian Process model.
- **Meta Bayesian Optimization (Meta BO) (Wang et al., 2018a)** - This approach meta-learns a Gaussian process prior offline over previously sampled data.
- **Learning Active Learning (LAL) (Konyushkova et al., 2017)** - This approach meta-learns an acquisition function leveraging hand-engineered features.

We benchmark against each of the above algorithms in both the ADMETS and high-dimensional domains. We do not compare to MAML (Finn et al., 2017) or Nagabandi et al. (2019) as these algorithms have no notion of safety or active learning.

## 5 RESULTS

We empirically validate that in the our meta-active learning approach outperforms baselines across both the ADMETS and high-dimensional domains in terms of its ability to actively learn latent parameters and its ability to safely perform these tasks. In contrast to our approach, even though BaO, Meta BO, and LAL do not consider safety, we are able to outperform those baselines across all metrics.

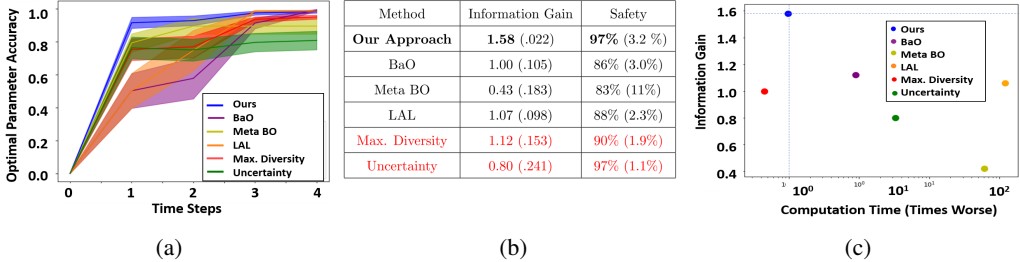

|  | (a) | (b) | (c) |

Figure 4: This figure depicts our empirical validation in the ADMETS domain, benchmarking algorithm accuracy per time step (Fig. 4a), overall (Fig. 4b), and vs. computation time (Fig. 4c).

**Active Learning –** Results from both the ADMETS and the high-dimensional domains empirically validate that our algorithm more efficiently and safely learns the optimal control parameters (Fig. 4) and system dynamics (Fig. 5). In Fig. 4b-5b, we report the mean (standard deviation) for each measure and baseline where possible. In the ADMETS domain, our model selects an action which results in a 58% higher information gain than BaO and 87% higher information gain than Meta BO on average. When compared to the state-of-the-art active learning baselines, our method performs 41% better than Maximizing Diversity and 98% better than Uncertainty in terms of information gain.

In the high-dimensional domain, we achieve a 49% improvement over Hastie et al. (2017) and a 46% improvement over both Schrum & Gombolay (2020) and Konyushkova et al. (2017). We point out that our approach outperforms active learning heuristics (i.e., Maximizing Diversity and Epistemic Uncertainty) when run inside our chance-constrained framework for a direct comparison. This result validates meta-learning an acquisition function is a necessary and beneficial component of our framework.

**Computation Time –** We demonstrate that our method also outperforms previous work in terms of computation time. Across both domains, our approach not only achieves a more efficient reduction in model error and improvement in information gain, we are also faster than all baselines in the more challenging, high-dimensional (Fig. 5c) environment. In the lower-dimensional, ADMETS

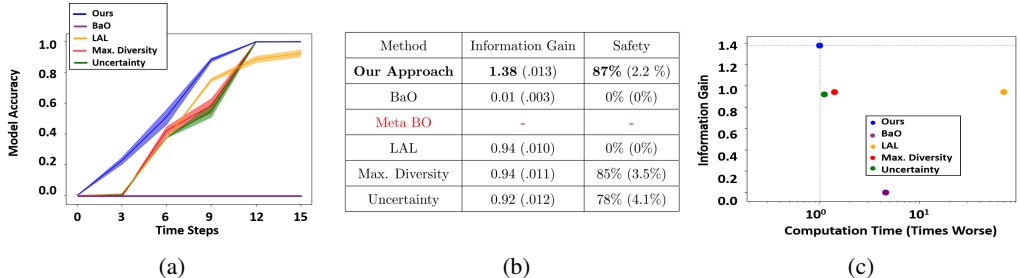

|  | (a) | (b) | (c) |

Figure 5: This figure depicts our empirical validation in the high-dimensional domain benchmarking algorithm accuracy per time step (Fig. 5a), overall (Fig. 5b), and vs. computation time (Fig. 5c). Error is calculated in batches of three time steps, enabling the robot to deviate from the safe region temporarily to gain information. Our algorithm outperforms baselines in both safety and information gain. We do not report full results for MetaBO in the high-dimensional domain as the high dimensional state space made this baseline computationally intractable.

environment (Fig. 4c), BaO has a slight advantage in computation time, but our algorithm trades the time for a 58% information gain over BaO. Additionally, we are 68x faster than LAL and 61x faster than Meta BO, the two other meta-learning approaches we benchmark against.

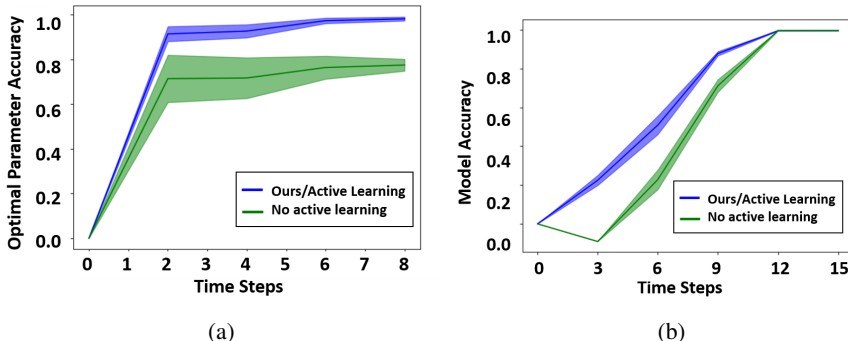

(a)                                                          (b)

Figure 6: This figure shows the results of our ablation analysis. In the above plots, we set $\lambda = 0$, meaning there is no active learning and only safety is maximized in our objective function. Fig. 6a shows the results in the ADMETS domain and Fig. 5b the results in the high-dimensional domain. In both domains, our meta-learned acquisition function is clearly an important component to achieve efficient learning.

**Safety –** In the high-dimensional domain, we empirically validate that, for the information results we report in Fig. 5 we can achieve an 87% probability the aircraft will return to the safe region 5a. As shown in Fig 5b, only Maximizing Diversity (85% safe) and Epistemic Uncertainty (78% safe) allow for safety constraints to be imposed. Our algorithm thus outperforms the baselines in safety as well as information gain. When we maximize for safety (i.e., $\lambda = 0$), our algorithm is able to achieve a 99.9% safe return to the hyper-ellipsoid even without a ground-truth dynamics model.

In the ADMETS domain, we find that our algorithm achieves a 2.3% higher guarantee of safety compared to Maximizing Diversity. Our baseline Uncertainty achieves an equivalent safety guarantee to our algorithm, yet our algorithm achieves a 98% greater information gain comparatively, placing it at the Pareto front in terms of safety and information gain. We outperform LAL, BaO, and MetaBo in information gain and computation time under our safety constraints even though these algorithms are unconstrained.

We show in Fig 6 the results of our algorithm in both domains when maximizing only safety and removing the active learning component from our objective. Without the meta-learned active learning function, our algorithm does not explore as efficiently and therefore achieves a lower model accuracy at each time step, demonstrating that our meta-learned acquisition function is an important component of our architecture.

## 5.1 DISCUSSION

Through our empirical investigation, we have demonstrated that our meta-learned acquisition function operating within a chance-constrained optimization framework outperforms prior work in active learning, meta-learning, and Bayesian optimization (Hastie et al., 2017; Schrum & Gombolay, 2020; Ashmaig et al., 2018; Konyushkova et al., 2017; Wang et al., 2018a). Specifically, we are able to simultaneously achieve an improvement in information gain via increased sample efficiency and decreased computation time. We achieve a 46% increase in information gain while still achieving a 20% speedup in computation compared to active learning baselines and 60x faster computation time compared to our meta learning baseline. Our novel, deep learning architecture, demonstrates a unique ability to leverage the power of feature learning across time-series data within a LSTM neural network and the utility of deep Q-learning, within mathematical optimization with chance constraints to explicitly tradeoff safety and active learning. Our approach allows for minimum safety guarantees while also maximizing the information gained based on a learned representation of sample history.

## 6   RELATED WORK

Our work lies at the crossroads of active learning, meta-learning and safe learning. We discuss the contributions of our work and why our approach is novel in light of prior work.

**Active Learning -** Active learning acquisition functions provide heuristics to selecting the candidate unlabeled training data sample that, if the label were known, would provide the most information to the model being learned (Burbidge et al., 2007; Hasenjager & Ritter, 1998; Cai et al., 2017; Hastie et al., 2017). For example, in Hastie et al. (2017) the action is selected that the learner is least certain about. In work by Ashmaig et al. (2018), the authors utilize the acquisition function Expected Improvement (EI) to balance exploration versus exploitation to determine the optimal stimulation parameters in ADMETS.

Prior literature has also investigated on-the-fly active learning and meta-active learning (Bachman et al., 2016; Konyushkova et al., 2017). Konyushkova et al. (2017) describes the algorithm Learning Active Learning (LAL). The authors present a meta-learning method for learning an acquisition function in which a regressor is trained to predict the reduction in model error of candidate samples via hand engineered features. Wang et al. (2018a) alternatively considers a Gaussian Process based method to meta-train an acquisition function on a distribution of tasks. They show that their method is capable of extracting structural properties of the objective function for improved data-efficiency. Work by Geifman & El-Yaniv (2018) attempts to actively learn the neural network architecture that is most appropriate for a given task, e.g. active learning. Pang et al. (2018) additionally proposed a method to learn an acquisition function that generalizes to a variety of classification tasks. Yet, this work has only been demonstrated for classification.

**Meta-Learning for Dynamics -** Prior work has attempted to address the problem of learning altered dynamics via a meta-learning (Clavera et al., 2009). Belkhale et al. (2020) investigate a meta-learning approach to learn the altered dynamics of an aircraft carrying a payload; the authors train a neural network on prior data to predict environmental and task factors to inform how to adapt to new payloads. Finn et al. (2018) present a meta-learning approach to quickly learning a control policy. In this approach, a distribution over prior model parameters that are most conducive to learning the new dynamics is meta-learned offline. While this approach provides fast policies for learning new dynamics, it does not explicitly reason about sample efficiency or safety.

**Safe Learning -** Prior work has investigated safe learning in the context of safe Bayesian optimization and safe reinforcement learning. For example, Sui et al. (2015) develop the algorithm SafeOpt which balances exploration and exploitation to learn an unknown function; however, this approach has significant limiting assumptions about the underlying nature of the task. Turchetta et al. (2016) address the problem of safely exploring an MDP by defining an a priori unknown safety constraint updated during exploration, and Zimmer et al. (2018) utilizes a Gaussian process for safely learning time series data. However, these approaches do not incorporate knowledge from prior data to increase sample efficiency, limiting their ability to choose the optimal action. Schrum & Gombolay (2020) attempt to overcome this problem by employing a novel acquisition function, Maximizing Diversity, which is utilized to quickly learn altered system dynamics in a chance constrained framework. Yet, the hand engineered acquisition function limits the capabilities of this approach.

## 7   CONCLUSION

In this paper we demonstrate a state of the art meta-learning approach to active learning for control. By encoding the context of the dynamics via an LSTM and learning a Q-function – which we encode into a mixed-integer optimization framework – that predicts the expected information gain of taking a given action, our framework is able to efficiently and safely learn the nature of the altered dynamics. We compare our approach to baseline acquisition functions and demonstrate that ours is superior in both computation time and information gain, achieving a 46% increase in information gain while still achieving a 20% speedup in computation time compared to state-of-the-art acquisition functions and more than a 58% higher information compared to Bayesian approaches.

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

# A APPENDIX

## A.1 DETAILS ON DOMAINS

Here we describe in more detail the input and output spaces of the ADMETS and high-dimensional domains as well as other details relevant to implementation.

### A.1.1 AIRCRAFT

Our simulation consists of ten states, i.e., forward velocity ($u$), vertical velocity ($w$), pitch rate ($q$), pitch angle ($\theta$), sideslip angle ($\beta$), roll rate ($p$), yaw rate ($r$), roll angle ($\psi$), yaw angle ($\phi$), and altitude ($Z$). The control inputs are elevator ($\Delta_e$), thrust ($\Delta^{(t)}$), aileron ($\Delta_a$), and rudder ($\Delta_r$). Our sampling rate is 20 Hz.

### A.1.2 ADMETS

The input parameter space is one-dimensional and consists of the voltage amplitude ($v_a$). The output space is a score quantifying memory, referred to as the discrimination score ($d_s$). The objective of this domain is to maximize discrimination area, which is defined as $d_a := d_s * v_a$. More detail on this domain and the data on which the simulation is based can be found at Ashmaig et al. (2018).

## A.2 DEFINITION OF INFORMATION GAIN

We define information gain, $I$, at time step $t$ as the percent decrease in the error of the objective as described in Eq. 3. $e^{(t)}$ is the error of the objective at time step $t$. In the high-dimensional domain, $e^{(t)}$ is defined as the mean squared error of the model, $e^{(t)} = \frac{1}{N} \sum_{i-1}^{N} (x^{(i)} - x^{\hat{(i)}})^2$. $x$ is the ground truth state and $\hat{x}$ is the state predicted by the model. In the ADMETS domain $e^{(t)}$ is defined as the $L_1$ norm of the predicted optimal parameter and the ground truth optimal parameter ($e^{(t)} = ||d_a - \hat{d}_a||_1$). During offline training, the ground truth can be obtained from the known model.

$$I^{(t+1)} = \frac{e^{(t)} - e^{(t+1)}}{e^{(t)}} \tag{3}$$

## A.3 BASELINE ACQUISITION FUNCTIONS

### A.3.1 MAXIMIZING DIVERSITY

We compare our method to the acquisition function, maximizing diversity, presented by (Schrum & Gombolay, 2020). Here, $u^{(i)}$ and $x^{(i)}$ are states and actions that the robot has previously experienced, and $f$ is the current dynamics model.

$$u^* = \underset{u \in \mathbb{U}}{\operatorname{argmax}} \sum_{i=1}^{N} \left\| u - u^{(i)} \right\|_1 + \beta \left\| \hat{T}_\psi(x, u) - x^{(i)} \right\|_1 \tag{4}$$

### A.3.2 MAXIMIZING UNCERTAINTY

This active learning metric described by (Hastie et al., 2017) quantifies the uncertainty in the output of the model for each training example as described in Eq. 5. Here, $\hat{T}_z(u)$) is the $z^{th}$ dynamics model and $\bar{T}$ is the average across models $z$.

$$u^* = \underset{u \in \mathbb{U}}{\operatorname{argmax}} \frac{1}{Z} \sum_{z=1}^{Z} \left\| \bar{T} - \hat{T}_z(u)) \right\|_1 \tag{5}$$

## A.4 ADDITIONAL DETAILS ON MIXED-INTEGER LINEAR PROGRAM FORMULATION

In this section we provide additional details on the linearization of our probability constraints and objective function for integration into a linear programming formulation.

### A.4.1 DYNAMICS MODEL REPRESENTATION

In practice, in the high-dimensional domain we find that $\hat{T}_\psi$ can be represented as a single-layer perceptron (i.e. linear regression) which advantageously is computationally efficient in domains that require fast computation times. We adopt a multi-layer perceptron with ReLU activations in the ADMETS domain. Our inferred dynamics therefore evolve according $\Gamma$. $A$ describes the evolution of a state with no input, $B$ the change in the state due to an action at time t, $\vec{u}^{(t)}$, and $I$ is the identity matrix.

$$\vec{X}^{(t+1:t+T)} = \beta\vec{x}^{(t)} + \Gamma\vec{\mathcal{U}}^{(t:t+T)} \tag{6}$$

$$\beta = \begin{bmatrix} A^2 & A^3 & \dots & A^T \end{bmatrix} \tag{7}$$

$$\Gamma = \begin{bmatrix} AB & B & 0 & 0 & \dots & 0 \\ A^2B & AB & B & 0 & \dots & 0 \\ \vdots & \ddots & & & & \\ A^{T-1}B & A^{T-2}B & A^{T-3} & \dots & \dots & B \end{bmatrix} \tag{8}$$

### A.4.2 LINEARIZATION OF PROBABILITY CONSTRAINTS

To include our safety constraints in a mixed-integer linear programming formulation, we remove the non-linearities via conservative assumptions and other techniques. Our safety constraints are defined in Eq. 9 and the dynamics evolve according to equations 6-8. $d$ represents the $d^{th}$ row and $j$ represents the columns.

$$\left\| \Phi^{-1}(1-\epsilon_d)\sqrt{\sum_j \sigma_{d,j}^2 x_j^{(t)^2} + \sum_j \sigma_{d,j}^2 \mathcal{U}_j^{(t:t+T)^2}} + \Gamma\vec{\mathcal{U}}^{(t:t+T)^2} - \Delta_d^{(t:t+T)} \right\|_1 < r_d \tag{9}$$

The following conservative assumption is made to linearize the sum of squares in Eq. 9.

$$0 \leq \sqrt{\sum_j \sigma_{d,j}^2 x_j^2} \leq \sum_j \sigma_{d,j}|x_j| \tag{10}$$

We utilize the binary decision variable, $\delta_\epsilon \in \{0,1\}$ as a probability selector variable to linearize the absolute value in Eq. 9. $M$ represents a large positive number and $\bar{\Gamma}$ is the point estimate of the dynamics. This linearization technique combined with the conservative assumption in Eq. 10 results in the following linear equations (Eq. 11-13) which can be integrated into a mixed-integer linear programming formulation. $E$ is the set of "probability levels", e.g., $E = \{0.75, 0.8, ...\}$ where $\min E$ defines the minimum enforced probability of safety.

$$-M\delta_\epsilon - \Phi^{-1}(1-\epsilon_p)\sum_j \sigma_{d,j}\tilde{\mathcal{U}}_j^{(t:t+T)} - \bar{\Gamma}\vec{\mathcal{U}}^{(t:t+T)} < \vec{r_d} + \Delta_d^{(t:t+T)} + \sum_j \sigma_{d,j}|x_j^{(t)}| \tag{11}$$

$$-M\delta_\epsilon + \Phi^{-1}(1-\epsilon_p)\sum_j \sigma_{d,j}\tilde{\mathcal{U}}_j^{(t:t+T)} + \bar{\Gamma}\vec{\mathcal{U}}^{(t:t+T)} < r_d - \Delta_d^{(t:t+T)} - \sum_j \sigma_{d,j}|x_j^{(t)}| \tag{12}$$

$$\sum_{p \in E} \delta_{\epsilon_{p,d}} = |E| - 1, \forall d \in D \tag{13}$$

### A.4.3 VARIANCE ESTIMATION

We compute uncertainty of our network via bootstrapping. We follow the method proposed in Efron (2020) and randomly redraw bootstrap training samples with replacement from our set of training data. This technique has been verified by Franke & Nuemann (1998) to be an effective method for

approximating the uncertainty of neural networks. The components of $\sigma$ are calculated according to Eq. 14. Here $\bar{x}$ is the average of the bootstrapped networks and $x_{d,j}^{(b)}$ a single bootstrapped network. $B$ is the number of bootstrapped networks.

$$\sigma_{d,j} = \sqrt{\frac{\sum_{b=1}^{B}(\bar{x}_{d,j} - x_{d,j}^{(b)})^2}{B-1}} \tag{14}$$

### A.4.4 LINEARIZATION OF Q-FUNCTION

Our Q-function includes a non-linear relu activation function which is linearized to be included in the mixed-integer linear programming formulation. Equations 15-17 define the equations for a neural network with relu activation. $\xi = \left[\left[\mathcal{U}^{(t:T)}\right]^{\mathsf{T}}, \left[\vec{Z}\right]^{\mathsf{T}}\right]$ is the input to the Q-function and $^{(l)}\omega_{j,i}$ is the connection between neurons $j$ and $i$ between layers $l$ and $l+1$.

$$Q(\vec{\mathcal{U}}^{(t:T)}, \vec{Z}) = \sum_{j} {}^{(2)}\omega_{j,d} {}^{(2)}o_j, \forall d \in D \tag{15}$$

$$^{(2)}o_i = \sum_{j} {}^{(1)}\omega_{j,i} {}^{(1)}o_j \mathbb{1}_{\left({}^{(1)}o_j \geq 0\right)}, \forall i \tag{16}$$

$$^{(1)}o_i = \sum_{j} {}^{(0)}\omega_{j,i} \xi_j^{(t)} \tag{17}$$

This formulation is linearized in Eq. 18-20. $k_i \in \{0,1\}$ is a binary indicator variable and $M$ represents a large positive number.

$$Mk_i - M + \sum_{j} {}^{(0)}\omega_{j,i}\xi_j^{(t)} \leq 0 \leq Mk_i + \sum_{j} {}^{(0)}\omega_{j,i}\xi_j^{(t)} \tag{18}$$

$$\sum_{j} {}^{(0)}\omega_{j,i}\xi_j^{(t)} - M \leq {}^{(1)}o_i \leq \sum_{j} {}^{(0)}\omega_{j,i}\xi_j^{(t)} + Mk_i \tag{19}$$

$$M - Mk_i \geq {}^{(1)}o_i \geq 0, \forall i \tag{20}$$

### A.4.5 LINEARIZATION OF "RESETTING" TERM

In practice, we find that taking a final, "resetting" action at time $t+T$ by adding $z_3 = \left\|\vec{x}^{(T)} - \vec{x}_r\right\|$ to minimize the distance between the aircraft's state and $\vec{x^r}$ helps to ensure the aircraft does not loiter along the boundary of safe operation until where a random perturbation could result in failure the aircraft. We linearize the resetting term in our objective function, i.e. the difference between our designated safe state, $\vec{x}_r$ and the final state $x_T$ by maximizing $-(z^+ + z^-)$ subject to the constraint in Eq. 21. $z^+$ and $z^-$ are both positive continuous variables.

$$\vec{x}_T - \vec{x}_r = z^+ - z^- \tag{21}$$
$$z^+, z^- > 0 \tag{22}$$

### A.4.6 LINEARIZED OBJECTIVE

The resultant linearized objective is defined in Eq. 23. In the ADMETS domain $\lambda_3$ is set to 0.

$$\vec{\mathcal{U}}^{(t:T)^*} = \operatorname*{argmax}_{\vec{\mathcal{U}}^{(t:T)} \in \vec{\mathcal{U}}^{(t:T)}} \left(\lambda_1 \sum_{j}^{H} ({}^{(2)}\omega_{j,d} * {}^{(2)}o_j - \bar{Q}_\theta(\cdot, \vec{Z}))\right.$$
$$\left. + \lambda_2 \sum_{p \in E} (1 - \delta_{\epsilon_{p,d}})\epsilon_{p,d} - \lambda_3(z_d^- + z_d^+)\right) \tag{23}$$

## B  SAFETY

### B.0.1  ADMETS DOMAIN

The baselines used in the ADMETS domain did not have built in safety guarantees. Therefore, when comparing against these baselines, we removed the safety constraints in our algorithm to make the comparison fair.

### B.0.2  HIGH-DIMENSIONAL DOMAIN

Safety constraints are necessary in the high-dimensional domain because without these constraints, the aircraft would quickly go into an unrecoverable configuration. The epistemic uncertainty Hastie et al. (2017) and diversity Schrum & Gombolay (2020) baselines can be integrated into our safety framework by linearizing the acquisition functions. Therefore, in our comparison of our algorithm versus the baselines, we enforced the safety constraints.

LAL Konyushkova et al. (2017), however, is not an inherently safe method of active learning. To fairly compare this baseline to our method in the high-dimensional domain, we simulate the possible actions that can be selected by LAL and discard those that are not safe (i.e., the actions that take the aircraft out of the cylinder of safety). Therefore, LAL can only select an action considered safe by our definition of safety.

### B.1  SENSITIVITY ANALYSIS

To robustly evaluate our method compared to the baselines, we vary the hyperparameters of the approach by (Schrum & Gombolay, 2020) for maximizing diversity and our meta-learned function to show that our function is robust and is still superior despite changes in hyperparameters. The results of this hyperparameter sweep are shown in Table 1. The hyper-parameter we vary for maximizing diversity is the number of previous training samples that we compare to. The information gain increases as the number of samples increases up to a point at which the selected sample tends to converge to the mean of the previously collected samples, causing the information gain to fall. We vary the number of hidden neurons in our Q-function as the hyper-parameter of interest as it governs the trade-off between computational speed and function approximation power.

In our approach, the addition of a hidden neuron adds an additional integer variable, resulting in an increase in computation time as demonstrated in Table 1. However, an addition of 25 neurons only increase the computation time by 3.7% and provides a 50% increase in information gain. In comparison, Schrum & Gombolay (2020)'s approach results in increase in information gain 4% while trading off a 6% loss in computation time. Likewise, increasing the number of bootstrapped models in Hastie et al. (2017)'s approach results in a 35% increase in information gain and a 37.5% increase in computation time. Therefore, in our approach we are able to gain more information without large increases in computation time.

| Parameter Setting | Our Approach | | Diversity (Schrum & Gombolay, 2020) | | | | | Uncertainty (Hastie et al., 2017) | | |
|---|---|---|---|---|---|---|---|---|---|---|
| | 5 | 30 | 1 | 2 | 3 | 4 | 5 | 2 | 3 | 4 |
| Average Information Gain (SD) | .196 (.37) | **.39** (**.23**) | .29 (.44) | .30 (.30) | .31 (.30) | .26 (.22) | .26 (.22) | .22 (.45) | .25 (.17) | .33 (.17) |
| Average Computation Time (s) (SD) | .12 (.03) | **.146** (**.01**) | .17 (.03) | .18 (.04) | .19 (.04) | .21 (.04) | 0.22 (.03) | .16 (.04) | .21 (.05) | .26 (.05) |

Table 1: Average information gain for our approach compared to that by (Schrum & Gombolay, 2020) and (Hastie et al., 2017). We vary the number of previous samples in the diversity maximization problem from one to five and the number of bootstrapped models from two to four in maximizing uncertainty heuristic. We vary the number of hidden neurons in our meta-learned Q-function. We **bold** the setting of our algorithm that outperforms our baselines across *all* hyperparameter settings tested.

## B.2 ADDITIONAL RESULTS

We present results for each damage condition in the high-dimensional domain comparing our approach to the baseline acquisition functions diversity Schrum & Gombolay (2020) and epistemic uncertainty Hastie et al. (2017). Our approach outperforms the baselines for each damage condition.

|  | Ours | Diversity | Uncertainty |
|---|---|---|---|
| Loss of Stabilizer | **.371** (.18) | .207 (.11) | .189 (.09) |
| Wing Damage | **.223** (.14) | .133 (.11) | .200 (.07) |
| Loss of Aileron | **.380** (.12) | .215 (.07) | .312 (.005) |

Table 2: We report average information gain and standard deviation for the canonical altered dynamics conditions loss of vertical stabilizer, wing damage, and loss of aileron control for the best parameter settings for each approach. Our approach outperforms both heuristics.

## B.3 HYPERPARAMETERS

The hyperparameters employed for each domain are listed in the Table 3. We show the learning rates for each domain which were determined via experimentation to be effective values. The size of the LSTM hidden layer and each of the two layers of the Q-function is also presented. $\tau$ is the soft update coefficient for updating the target network.

|  | ADMETS | Aircraft |
|---|---|---|
| Learning Rate | 1e-4 | 1e-3 |
| LSTM Hidden Layer Size | 20 | 100 |
| Q function Layer 1 Size | 64 | 50 |
| Q function Layer 2 Size | 16 | 30 |
| Soft Update ($\tau$) | .001 | .001 |
| Exploration noise | Gaussian | Ornstein-Uhlenbeck |

Table 3: Hyperparameters for the ADMETS and high-dimensional domains.

