# OpenReview forum: "Meta-Active Learning in Probabilistically-Safe Optimization"
_ICLR.cc/2021/Conference — Reject_

### Official Review · AnonReviewer2 · 2020-10-28
**Experiment section is somewhat problematic**

**Rating:** 6
**Confidence:** 3

**Review:**

The authors proposed a meta-learning approach for active learning in the context of robot control.

What I like about this work is that there're several illustrations that make it clear which problem setup it is or what the authors meant by safety. The authors also included several competing methods, and the experiments showed promising results both in terms of several metrics, including information gain and safety.

However, I also have some concerns about this paper that needs to be addressed.

In the experiments, there are 5 methods chosen to be compared against. However, several of them are not even mentioned before or after in introduction or related work to explain their close connections to the proposed one in more details. For example, Schrum & Gombolay 2020 and Wang et al., 2018b. And that makes me doubt whether it's a poor selection of comparisons or omission of important details in writing.

Again in experiments, the method names described in 4.3 don't match the ones shown in figures. What's Random Bayesian in Figure 4? Why it disappeared in Figure 4(b) and 4(c)? Where's Epistemic Uncertainty and Maximizing Diversity? Similar for Figure 5, it was said that only Meta BO is excluded but where are the others? And even for Meta BO, the paper says it is "not included as it was designed for optimization tasks". But why can't the task in sec 4.2 be framed as an optimization task? What I meant is, I would like to see consistent set of methods in different tasks. Or add more tasks to show generality.

There are also inconsistency in naming. Is it BaO or Bao?

About the safety measure. It's evaluating using a point, and all parameters like minimum risk value need to be set by hand. How useful is such a safety measure? And how is it related to more common ways of measuring safety/risks, e.g. CVaR? Also the variables in Eq. (2) are not fully explained.

I'm also somewhat concerned that the authors are trying to solve too many problems in one algorithm. Safety, meta-learning, Q-learning.. But the experiments only show the results of everything combined without detailed analysis of the performance of each functionality while keeping others fixed. To some extent, the nature of the design of the method also sort of forbids modular tests. In the end, it might be just another impractical approach that incrementally adds several factors together, yet it's impossible to explain when it works or why it works.

---

> ### Author Response · Authors · 2020-11-14
> **Response**
>
> We would like to thank the reviewer for their reviews and their comments about our illustrations.  For the reviewer’s convenience, we have highlighted all changes in red in the updated paper newly attached. Below we address the concerns expressed in the review.
>
> *[R2] Benchmarks are not mentioned in the introduction or related works.*
> We thank the reviewer for their comment, and we now include a discussion of these methods in the related works to more properly place them in context. The additions have been highlighted in red.
>
> *[R2] Not all benchmarks are included in each environment.*
> We agree with the reviewer that we should have demonstrated each benchmark in both environments. We have included these benchmarks in both environments as reflected in the revised version of our paper. We have updated the figures and results to make this clearer and have eliminated any inconsistencies with the benchmarks.
>
> *[R2] There are inconsistencies in naming*
> We thank the reviewer for pointing this out and we have corrected our naming conventions so that they are consistent throughout the paper.
>
> *[R2] The safety measure must be set by hand. How useful is such a measure? How does this compare to CVaR?*
> There are two parameters that must be set by hand: A) the minimum-safety level allowed  (i.e., $p_{min}= min_{p’ \in E}  p’ $, as described in the sentence preceding Equation 11), and B) the safety vs. active learning trade-off parameter, \lambda. We agree that other measures may be interesting to consider. In the case of CVaR; however, it is nonobvious how one would perform the necessary quadrature over the product of the weighted probability distribution within a linear programming formulation. We appreciate the reviewer’s suggestion and will consider value-based and other measures in future work.
>
> *[R2] Variables in Eq. 2 are not fully explained.*
> We thank the reviewer for their feedback and have corrected this oversight.
>
> *[R2] The authors are trying to solve too many problems at once.*
> We disagree that we are trying to solve too many problems. We developed a novel approach to active learning that meta-learns an acquisition function while also allowing for safety constraints to be imposed. We argue that each of these components is a necessary part of our algorithm. Safe active learning is highly useful and has many applications in safety-critical domains. While our algorithm can operate without safety constraints, this would be highly dangerous in the aircraft or ADMETS domain in which safety is a critical requirement. Additionally, we compare our algorithm’s meta-learned acquisition function against two state-of-the-art hand engineered acquisition functions. We show that our algorithm outperforms these approaches, demonstrating that meta-learning is a necessary component for efficient learning.
>
> To address the reviewers concerns about a lack of modular tests, we include an ablation analysis in which we remove the active learning component in our algorithm (i.e., our algorithm only maximizes for safety) and show that the active learning component of our algorithm is crucial to efficient learning.
>
> We hope this response addresses your concerns. Please let us know if you have any further comments or concerns or ways in which we can further improve our paper

---

> > ### Comment · AnonReviewer2 · 2020-11-23
> > **Reply to response**
> >
> > I have read the author response and roughly went through the revised paper again. I appreciate that the authors addressed the problems pointed out and I plan to increase the score to 6. The reason that I probably won't increase further is that I still think this paper is solving too many problems all together without a clear focus or significant intellectual contribution. What I hope to see is deep dive into one topic, e.g. meta learning for acquisition function, and teach readers this subject thoroughly. Or alternatively, if it's really important to put all these together, I think it's important to give enough motivation why everyone is needed, e.g. detailed description of specific tasks in intro and why they are important.

---

> > > ### Author Response · Authors · 2020-11-24
> > > **Thank you**
> > >
> > > We would like to thank the reviewer for the reviewer's insightful comments and helpful feedback in improving our paper. Thank you!

---

### Official Review · AnonReviewer3 · 2020-10-28
**the paper is well-motivated and the proposed approach performs well empirically, while the paper is more about a specific application and somehow out of the scope of ICLR**

**Rating:** 5
**Confidence:** 3

**Review:**

#### Paper summary:

In this work, a meta active learning approach is proposed to learn the hidden dynamics of control systems, where safety is also a major concern. The main idea lies in doing meta-learning with Q-learning, meanwhile selecting safe actions by solving a mixed-integer linear programming problem. The performance of the proposed approach is verified from real datasets of deep brain stimulation by outperforming existing baselines for a significant gap in both accuracy and computational complexity.

#### Advantages:

- The paper discusses a very meaningful problem. Active learning with safety constraints is very important in many applicational domains.

- From the experiments, the proposed approach shows very significant performance in both the deep brain stimulation and the dynamics recovering tasks.

#### Disadvantages:

- From the perspective of machine learning, the proposed approach is a combination of existing techniques, thus the technical novelty is limited.

- The paper is more about solving a specific robotics problem, which may not lie in the major focus of the ICLR conference.

Overall evaluation:

On one hand, I think the paper discusses a very meaningful problem, while on the other hand, I think ICLR may not be the most suitable venue to publish this work.

---

> ### Author Response · Authors · 2020-11-14
> **Response**
>
> We would like to thank the reviewer for their time invested in our paper and for their comments about the importance of our work in safety critical domains. For your convenience, we have highlighted all changes in red in the updated paper. Below we address the concerns expressed in the review.
>
> *[R3] The approach is a combination of existing techniques, thus the technical novelty is limited*
> Virtually every paper published at ICLR this year will be a “combination” of one or more techniques. The vast majority of research consists of finding novel connections between different techniques that will allow one to create something that is greater than the sum of its parts. Yes, we employ LSTM models, chance-constrained programming, and Bellman’s Equation, which have all been established and well-studied. Nonetheless, to the best of our knowledge, our paper is the very first to show that an LSTM can encode a powerful representation of an active learning sampling history that can be directly leveraged in a chance-constrained mixed-integer program to optimize for information gain vs. safety. Furthermore, ours is the first to show than an acquisition function can be meta-learned for the purpose of the optimization and identification of a latent system. We appreciate the reviewer’s feedback that the novelty could be more clearly articulated, and we now do so in Section I.
>
> *[R3] The paper is more about solving a robotics problem*
> We argue that our paper is about solving meta-active learning in a chance-constrained optimization framework. While we do demonstrate our algorithm in robotics domains and a healthcare domain, the primary focus of our paper is the development and validation of a novel, meta-active learning approach that achieves state-of-the-art results against machine learning and Bayesian optimization baselines.
>
> *[R3]  ICLR may not be the best venue*
> We argue that our work is well-suited to ICLR due to the fact that a critical component of our method is its ability to meta-learned a representation of sample history which can encoded in a mixed-integer linear program. We believe that our work fits nicely under the following relevant topics listed in the ICLR guidelines:
> •	unsupervised, semi-supervised, and supervised representation learning
> •	representation learning for planning and reinforcement learning
> •	learning representations of outputs or states
> •	applications in audio, speech, robotics, neuroscience, computational biology, or any other field
>
> We hope this response addresses your concerns. Please let us know if you have any further comments or concerns or ways in which we can further improve our paper

---

> > ### Comment · AnonReviewer3 · 2020-11-24
> > **reply to response**
> >
> > Thanks for the detailed reply. I think the paper did a good job in solving the specific learning problem, i.e. meta active learning the system dynamics for control with safety constraints. What I meant in my previous review is that in my view, a good ICLR paper should make some novel technical contributions in representation learning or other areas of machine learning. This means that there should some new algorithms or observations invented from the view of learning itself, not just to utilize some well-established techniques to solve a new problem. I think it is very useful to explain more of the novelty of the proposed approach from a technical perspective.

---

> > > ### Author Response · Authors · 2020-11-24
> > > **Reply to Reviewer 3**
> > >
> > > We appreciate the reviewer’s feedback and we would like to thank the reviewer for their reply. Our approach to learning a representation of a sampling history and state that can be encoded into a MILP and optimized over is entirely novel. It is a groundbreaking way to connect mixed-integer linear programming/symbolic optimization and Deep Learning and, to the best of our knowledge, has never been done before. This novel integration, importantly, allows us to impose chance-constraints and ensure system safety which is an advantage over previous approaches in safety critical domains. We believe that our learning representation opens up new and exciting opportunities to bridge connectionist and symbolic approaches.

---

### Official Review · AnonReviewer4 · 2020-10-28
**An interesting paper that combines LSTM and mixed integer programming for meta-active learning but suffers from poor writing**

**Rating:** 6
**Confidence:** 3

**Review:**

Summary:
This paper presents a meta-active learning approach to obtain an LSTM-based embedding of a dynamic system and use a chance-constrained (probabilistic safe) optimization to find optimal control configuration via applying mixed-inter linear programming (MILP) on the learned embeddings of the dynamic system. The main idea is to learn a Q-function as an acquisition function to describe the future information gain, which is the percent decrease in the error of the objective (e.g. model error) via a meta-learning strategy in which the agent interacts with the environment via distribution of altered dynamics.

Strength:
-  To my best knowledge, the idea that combines an LSTM embedding of dynamics with mixed-integer programming for optimization is novel meta-active learning algorithms.
Weakness:
- The algorithm relies on the assumption that the safety region must be hyper-ellipsoid with a known radius and the model error must come from a Gaussian distribution with known mean and variance. This makes the algorithm hard to apply for problems when the safety region is unknown or changing over time, which happens often in most real-world problems.
- The poor writings make it hard to access the correctness of the technical details.

Questions and comments:
1) In Algorithm 1 line 11, should the subscript “T” be “t”? should the index “i” starting at line 7 be “t” and all the related subscript “i” be “t” instead?
2) In Eq (1), should the subscript of U be “t:t+T” instead of “t:T”? If keeping “t:T”, the sentence above Eq (1) that states “the trajectory from [t, t+T)” should be “[t, T]”. Alternatively, the author could formally define what “t:T” means. Also, in the first paragraph of section 3.2, it’ll flow better if the authors could give some description of the second part on the right-hand side of Eq (1).
3) Eq (2) is a bit hard to understand. What is \Delta_d^{t:2}? How is it set in practice? On the fourth line of the first paragraph on page 5, the paper states “and \bar{a} and \bar{b} are the point estimates of dynamics”, what is “\bar{a}” and “\bar{b}”?  The next sentence “d represents the d-th row and j the columns” is confusing.
4) Eq (2), the first term under the square root has x_j^{(t)}^2, should the underscript “j” be “d’” instead?
4) In the ADMETS experiment, the objective is defined as the L1 norm of the predicted optimal parameter and the ground truth optimal parameter;  how is the ground truth optimal parameter obtained? If the ground truth optimal parameter is already available, what’s the need for running the algorithm since the goal is to find the optimal parameters of the configurations?
5) In the Aircraft experiment, the objective function is defined as the mean squared error of the model (see Appendix A.2), what is the model here? What is \hat{x}_i and x_i?
6) In Appendix (A.4.2), what is the E in equation 13?
7) Page 8, “Meta-learning for Dynamics”, line 6, “… most conducive …” should be “… most conductive …”, line 7, “While this approach provide .. “ should be “While this approach provides …”
8) Eq (20) becomes 0 >= \prescript {o_i}  >= 1 when k_i =1, this will never be satisfied.

---

> ### Author Response · Authors · 2020-11-14
> **Response**
>
> We would like to thank the reviewer for their reviews and for their time invested in the details of our paper. For your convenience, we have highlighted all changes in red in the updated paper. Below we address the concerns expressed in the review.
>
> *[R4] The algorithm relies on the assumption that the safety region must be a hyper-ellipsoid with a known radius and the model error must come from gaussian distribution with known mean and variance. This is hard to apply for real world problems in which region is unknown/changing*
> The mean and variance of the gaussian distribution are not known a priori and are estimated online. Additionally, we proffer that our definition of the safety hyper-ellipsoid is applicable to many, real-world problems. For example, in the situation of a damaged aircraft, one could also define this region simply as maintaining a minimum altitude above ground level, which would not be a time-varying parameter. In the ADMETS domain, the system remains safe if the memory score is greater than zero. This boundary is defined within the medical community and is a constant region which does not evolve. We do acknowledge that there are scenarios in which this safety region may change. However, estimating the safety region online is an open problem outside of the scope of this paper.
>
> *[R4] The poor writing makes it difficult to assess technical details*
> We appreciate the reviewer’s feedback and have sought to improve the writing of the paper. We took care to address all comments by the reviewer including clarification of symbols and equations. The changes are highlighted in red in our revised paper
>
> *[R4] Critiques 1-4,6,7,9*
> We agree with the reviewer’s critiques and have updated our paper accordingly. Thank you.
>
> *[R4] Critique 5: How are the ground truth optimal parameters obtained and what is the need for running the algorithm if they are already known?*
> We agree with the reviewer that the objective of our algorithm is to determine the ground truth parameters. Within our training data set, the ground truth parameters are only known and available for offline training. These ground truth parameters are obtained from the ground truth model in our simulation and are used to calculate the reward during meta-learning. However, during testing, these ground truth parameters are unavailable to the model, and the model infers a probability distribution over these parameters.
>
> *[R4] Critique 8: Conducive should be conductive*
> Based upon the definition of conductive (having the property of transferring heat or electricity), we believe that conducive is the proper choice.
>
> We hope this response addresses your concerns. Please let us know if you have any further comments or concerns or ways in which we can further improve our paper

---

> > ### Comment · AnonReviewer4 · 2020-11-23
> > **Follow Up**
> >
> > I think the authors did a great job of improving the notation of the equations. Some minor issues are remaining (I missed them at the original review):
> > 1) \Gamma_k , the underscript k is not explained.
> > 2) Since the $i$ in Algorithm 1 is changed to $t$, it'll be more consistent to change the iteration index $i$ in Algorithm 2 to $t$ as well.
> >
> >
> > About the assumption of the safety region being a Gaussian distribution, I agree with the authors that in a lot of practical applications, Gaussian distribution might be enough as long as the mean and the variance are estimated online.

---

> > > ### Author Response · Authors · 2020-11-23
> > > **Response to Reviewer 4 Follow Up**
> > >
> > > We have sought to carefully incorporate all of the reviewer’s feedback. Given the reviewer’s latest response that we have greatly improved our notation and addressed concerns about the assumptions of the safety region, we are wondering if the reviewer would consider raising the paper’s score or offering additional points for improvement?

---

> > > > ### Comment · AnonReviewer4 · 2020-11-24
> > > > **Response**
> > > >
> > > > I think the the clarity of the paper has been improved substantially at this point and would be happy to increase my score to 6. The writing could still use some improvement, which will help to better highlight the novelty of the paper.

---

> > > > > ### Author Response · Authors · 2020-11-24
> > > > > **Thank you**
> > > > >
> > > > > We would like to thank the reviewer for the reviewer's insightful comments and helpful feedback in improving our paper. Thank you!

---

### Official Review · AnonReviewer1 · 2020-11-01
**A Complex Framework that Deals with Safety-Critical Problems**

**Rating:** 5
**Confidence:** 2

**Review:**

The paper proposes a framework that learns the dynamic of safety-critical systems. The framework makes use of ideas from active learning, meta-learning, and reinforcement learning. As far as I understand, the framework seeks to learn an acquisition function that can evaluate the information gain from a certain action taken at a certain state. Acquisition function is a function that identifies an action the agent desires to take from a collection of unvisited states. The learning of such an acquisition function is formulated as a reinforcement learning problem, trained in a meta-learning fashion. I have the following comments:

1. The proposed method is evaluated on two problems, ADMETS and damaged aircraft.  Both problems are solved using synthetic data. While it is understandable that the framework is trained on simulated data, is there any justification with respect to the quality of the data simulator used in these two problems? Why performing well on these simulated data would be an indication of the proposed method can work reasonably well in practice?

2. Since the proposed framework is trained in a meta-learning fashion, is there any quantification on how fast/well (e.g. wrt. sample complexity) the trained policy adapting to new environments in test time?

3. While the paper takes safety during the learning of the dynamics as a critical issue, is there any justification with respect to the percentage of safety achieved by the proposed method? For example, why an 87% safety reported in Figure 5 achieved by the proposed method is a good number?

4. The key components of the proposed framework (e.g. meta-learning, reinforcement learning, active learning, safety) make sense to address the technical challenges in learning system dynamics in safety-critical domains. It is not very clear to me if the novelty of the proposed framework is appropriately highlighted. The paper can benefit from a better highlight of the novelty of the proposed method and why such novel contributions matter.

---

> ### Author Response · Authors · 2020-11-14
> **Response**
>
> We would like to thank the reviewer for their thorough review and insightful critiques. For your convenience, we have highlighted all changes in red in the updated paper. Below we address the concerns expressed in the review.
>
>
> *[R1] Critique 1: Is there any justification for the quality of the simulator and translation to the real world?*
> Our aircraft simulator leverages the equations of motions derived by Watkiss (1994) and Zhang et al (2017), which have been demonstrated to have high fidelity in real-world evaluations.  We believe the strong grounding in literature is a convincing and justifiable reason to utilize this simulation approach. The ADMETS domain is based on a simulation employed in the medical community and presented by Ashmaig et al. (2018).  Based on the facts that 1) the medical community is accepting of results demonstrated in this simulation and 2) simulation results have been shown by Ashmaig et al (2018) to transfer to the real world, we believe it is a justifiable simulation. We were unfortunately prevented due to COVID-19 from conducting an in vivo demonstration of our algorithm and we acknowledge that a lack of a real world demonstration is a limitation.
>
> *[R1] Critique 2: Is there quantification on how fast/well the trained policy adapts to new environments in test time?*
> Yes, Section 5 of our original submission included this result. Figures 4 and 5 show that our algorithm learns the objective faster than the baselines and outperforms baselines in terms of computational time. Please let us know if the reviewer has further questions about these results that we can clarify.
>
> *[R1] Critique 3: What is the justification for the safety achieved being a good value?*
> Our algorithm allows the user to choose a minimum level of safety. Our algorithm cannot return a solution that does not meet or exceed this safety standard. Above and beyond this minimum safety specification, our algorithm also seeks to maximize a dual-criteria objective to be even more safe if possible while seeking out additional information for active learning. For example, in the high-dimensional domain, we set the minimum safety threshold to 60%, yet our algorithm is able to guarantee at least 87% safety when maximizing for both safety and information gain.
>
> *[R1] Critique 4: The authors should further highlight the novelty of their contribution*
> Our novel computational method is the first to meta-learn an acquisition function for the purposes of system optimization. Additionally, to the best of our knowledge, ours is the only algorithm that can embed this meta-learned active learning function in a chance-constrained linear programming framework, allowing for the imposition of safety constraints. We more clearly outline the novelty of our contribution in Section I in the revised paper. We hope this response addresses your concerns. Please let us know if you have any further comments or concerns or ways in which we can further improve our paper.

---

> ### Author Response · Authors · 2020-11-24
> **Follow up**
>
> We hope that our responses to the reviewer’s critiques relieved the concerns the reviewer had. We sought to address the reviewer’s comments regarding the quality of our simulations, the ability to adapt to new environments, safety and novelty. We would like to ask the reviewer if there are any further clarifications/revisions they would like to see that might improve our score. Our discussion period ends today so this is our last opportunity to provide further clarifications and changes to our paper to satisfy the reviewer’s concerns

---

### Decision · Program_Chairs · 2021-01-07
**Final Decision**

**Decision:**

Reject

**Comment:**

This paper was quite contentious.  While reviewers appreciated the detailed response by the authors, and there is consensus that the paper addresses a relevant problem and contains interesting ideas, in the end there remain several concerns.  The paper provides a complex combination of techniques from active learning, meta learning and symbolic reasoning (via MILPs), and there are concerns about the clarity of the exposition.  For a paper claiming safety properties, there is also a lack of either formal theoretical analysis of well-specified safety properties, or a compelling demonstration of its effectiveness on a real system (all experiments are carried out in simulation).